# Cross-Domain Echocardiography Segmentation with Multi-Space Joint Adaptation

**DOI:** 10.3390/s23031479

**Published:** 2023-01-28

**Authors:** Tongwaner Chen, Menghua Xia, Yi Huang, Jing Jiao, Yuanyuan Wang

**Affiliations:** 1Department of Electronic Engineering, Fudan University, Shanghai 200433, China; 2Key Laboratory of Medical Imaging Computing and Computer Assisted Intervention of Shanghai, Shanghai 200032, China

**Keywords:** echocardiography segmentation, adversarial learning, domain adaptation

## Abstract

The segmentation of the left ventricle endocardium (LV_endo_) and the left ventricle epicardium (LV_epi_) in echocardiography plays an important role in clinical diagnosis. Recently, deep neural networks have been the most commonly used approach for echocardiography segmentation. However, the performance of a well-trained segmentation network may degrade in unseen domain datasets due to the distribution shift of the data. Adaptation algorithms can improve the generalization of deep neural networks to different domains. In this paper, we present a multi-space adaptation-segmentation-joint framework, named MACS, for cross-domain echocardiography segmentation. It adopts a generative adversarial architecture; the generator fulfills the segmentation task and the multi-space discriminators align the two domains on both the feature space and output space. We evaluated the MACS method on two echocardiography datasets from different medical centers and vendors, the publicly available CAMUS dataset and our self-acquired dataset. The experimental results indicated that the MACS could handle unseen domain datasets well, without requirements for manual annotations, and improve the generalization performance by 2.2% in the Dice metric.

## 1. Introduction

Cardiovascular disease (CVD) is the leading cause of death among adults worldwide. Echocardiography, the non-invasive imaging inspection tool, is widely applied in the clinical routine of CVDs [1]. The analysis of echocardiography is often used clinically for appraising cardiac morphology and function. For example, the ejection fraction (EF) [2] of the left ventricle (LV) is one of the famous clinical indices to quantify the LV systolic function. The LVEF can be calculated by the volume of the left ventricle. The width of the left ventricular myocardium also contains pathological information. The exact segmentation of the left ventricle endocardium (LV_endo_) and the left ventricle epicardium (LV_epi_) provides the clinical quantitative measures mentioned above. However, manual echocardiography labeling requires a doctor with rich clinical expertise to spend a lot of time, which seriously affects diagnostic efficiency. Smart health [3,4] can assist clinical diagnosis, such as automatic echocardiography segmentation. Automatic echocardiography segmentation remains an unresolved challenge due to the low signal-to-noise ratio and low contrast. The boundaries between anatomical structures are ambiguous and sometimes missing in echocardiography, and the performance of traditional segmentation methods [5,6] is not satisfactory. 

The development of deep learning [1,7] has promoted automatic echocardiography segmentation. The U-Net [8] has provided a good idea for medical image segmentation. Further, the RU-Net [9] merges two U-Nets as an attention mechanism to obtain the region of interest (ROI) in the image. The first U-Net is used for preprocessing to simplify the ultrasound image, and the second U-Net is corrected on this basis in order to produce the final segmentation result. Then, the Res-U [10] network couples the advantages of the ResNet and U-Net to denoise the ultrasound images. More recently, some methods have adopted echocardiography prior knowledge to help with LV segmentation. The anatomical constrained neural network (ACNN) [11] incorporates the prior knowledge of the heart anatomy structures to enhance the constraints on the segmentation boundaries. Veni et al. [12] propose a network that combines deep learning and a shape-driven deformable model for LV segmentation. Cui et al. [13] propose the multi-constraint aggregation learning (MCAL), which judges the boundary by anatomical knowledge and gives the boundary region a higher weight. Painchaud et al. [14] deform illogical results with a variational autoencoder that learns the cardiac shape. Ensemble learning also plays a role in echocardiography segmentation. The TRSA-Net [15] adopts an innovative co-attention learning structure for both segmentation and quantification tasks. The CLAS [16] achieves temporal-consistent segmentation on echocardiographic sequences with co-learning from the appearance and shape.

As a result of the diverse imaging devices and imaging protocols, echocardiography datasets are greatly different from each other in domain styles, such as grayscale distribution. Figure 1 shows an example of echocardiography from diverse centers. These images are inconsistent in the grayscale distribution and spatial texture. This discrepancy causes the segmentation network trained on one dataset (source domain) to not be able to exert effectively on another dataset (target domain) directly. This problem is expected to be mitigated via unsupervised domain adaptation, i.e., unpaired image-to-image translation. It causes one domain image to be visually similar to another domain image, while preserving the content information in the source images. Domain adaptation models based on the generative adversarial network (GAN) have been widely used in natural image processing. The SAGAN for image-to-image translation [17] introduces a self-attention network to the GAN and bidirectionally translates the style feature with the global information. The F-LSeSim [18] captures the spatial relationships of two domains and preserves the scene structure consistency via a new self-supervised learning method. The CUT [19] is a contrastive learning method for patch-based image-to-image translation. The F-LSeSim and CUT only learn the translation in one direction.

The above-mentioned domain adaptation methods perform well in natural images, but are not fully functional in medical images; this is because most medical images have low color contrast. It is difficult to maintain the key structure by the domain adaptation in the image space [20], and a reasonable approach is the domain adaptation in other spaces. Recently, several approaches have been tried in this way on optical medical images. The p-OSAL [21] is a patch-based output space adversarial learning network for segmenting optical medical images from different datasets. The BEAL [22] network uses two discriminators driven by the boundary and entropy in the output space. The IOSUDA [23], based on the BEAL, aligns different datasets in both the input and output space. There is also some related research on grey-scale medical images. Yan et al. [24] propose a multi-output adapter for cross-vendor left ventricle segmentation on cine MRI sequences. The SIFA [25] accomplishes synergistic image and feature alignment across CT and MR images. However, there has been no research on domain adaptation for echocardiography segmentation.

In this study, we present a multi-space adaptation-segmentation-joint network for extracting the LV_endo_ and LV_epi_ in echocardiography. The method is named the MACS. It adopts a GAN architecture, integrating the segmentation part (generator) and domain adaptation part (multi-space discriminators) to handle the cross-domain echocardiography analysis. The segmentation part utilizes an attention mechanism to extract the multi-scale features. Then, a dual-branch prediction module decodes the features into segmentation masks, with an emphasis on both the target region and boundary. The domain adaptation part aligns the samples from different domains in both the feature space and output space, enforcing generators to refine the segmentation results. 

Our main contributions lie in the following. The MACS achieves the segmentation of the LV_endo_ and LV_epi_ in multi-domain echocardiography datasets using labels of only one domain for training. Specifically, it adapts the source domain and the target domain on both the feature space and output space, which avoids the side-effect of direct translation on the original ultrasound images [23] and improves the segmentation accuracy. The joint adaptation on the feature space and output space promotes the model adaptation. Furthermore, detailed experiments were performed on two datasets from different medical centers and devices. The ablation studies illustrate the contributions of each module of the MACS. The comparisons with the state-of-the-art methods confirm that our method achieves accurate segmentation on the unseen domain dataset. To the best of the authors’ knowledge, it is the first model [1] for cross-domain echocardiography segmentation.

The paper is organized as follows. Section 2 details the proposed method. Section 3 describes the experiment details. The experimental results are reported in Section 4 to demonstrate the effectiveness of our method. In Section 5, we provide the extended discussion. The conclusion is given in Section 6.

## 2. Methods

The MACS framework is shown in Figure 2. It consists of a generator (an attention-based dual-branch segmentation module, ASDB) and multi-space discriminators. First, images and annotations from the source domain datasets are used to optimize the ASDB and obtain the intermediate features as well as the segmentation predictions. Entropy maps are further calculated from the segmentation predictions. Simultaneously, images from the target domain datasets also pass by the ASDB to generate the intermediate features and entropy maps. Then, a feature discriminator and an entropy discriminator distinguish the features and entropy maps from the different domains, respectively. The game between the generator and discriminators forces the distributions of the predictions on the target domain to fit that on the source domain. Thus, the MACS achieves accurate segmentation on both the labeled source domain and the unlabeled target domain.

### 2.1. Attention Segmentation with Dual-Branch(ASDB)

Semantic segmentation in ultrasound images is affected by ambiguous boundaries and speckle noise. The multi-scale attention mechanism [26,27] has been proven to perform well on ultrasound image segmentation. The ASDB (Figure 3) is inspired by the structure of the Deep Attention Feature (DAF) [26,27] and obtains both the global and local context from the features of the multiple levels. The layer-wise attention mechanism can assign the attention weight of individual layers to comprehensively optimize the segmentation model. The two branches of the ASDB are the output mask predictions ($P_{X}^{m}$) and boundary predictions ($P_{X}^{b}$), respectively. The dual-branch structure learns from the multi-scale features for the robust segmentation. 

The ASDB extracts four feature maps of different scales using the continuous ResNeXt. The low-level and high-level features contain detailed and semantic information, respectively. The layers of features are refined by the deep attention model (DAM). Specifically, the multi-level features are fused and reweighted to form the multi-layer feature (MLF). The MLF and single-layer feature maps are fed into the DAM for the refined feature maps. The mask branch combines the original feature maps and the refined feature maps from all levels and averages them for the final segmentation prediction. The boundary branch learns from the refined feature maps of the lowest two feature layers.

The loss function of the ASDB module is:(1)$${ℒ}_{seg}=\sum_{i=1}^{n}{ℒ}_{bce}^{i}+{ℒ}_{mse}^{b},$$
where ${ℒ}_{bce}^{i}$ represent the binary cross-entropy loss of the i-th layer prediction,$\;{ℒ}_{mse}$ is the mean square error (MSE) loss for the boundary regression.

### 2.2. Multi-Space Discriminators

Multi-space discriminators take the segmentation predictions of both the source and target domains as inputs and classify them according to their similarity of the distribution. The discrepancy between the echocardiographies from different domains mainly contains the grayscale distribution, the spatial texture, the ventricular location, the angle of probe scanning, etc. While the echocardiography may appear different, the feature and output of the segmentation share strong similarities (e.g., shape and topology). Therefore, adversarial learning on the feature space and output space encourages a narrowing of the distribution gap of the segmentation results between the source and target domain datasets. Then, the segmentation network overcomes the performance degradation caused by the domain migration and works on the target domain images.

#### 2.2.1. Discriminator on Feature Space

The well-trained segmentation without the domain adaptation tends to produce inaccurate predictions on target domain images. To obtain the semantic segmentation information from the global features extensively, we chose higher-level features, extracted by the segmentation network as inputs of the feature discriminator. As high-level features contain lots of contexts and semantic information, their alignment could force the encoded features of the source domain images and the target domain to be semantically consistent. 

The feature discriminator D^f^ aligns the high-level feature distributions by classifying whether the feature is produced from the source domain or the target domain. When the discriminator cannot judge from which domain features are extracted, the source domain and target domain are successfully aligned on the feature space. Otherwise, gradients of the discriminator D^f^ are delivered backward to encourage the features of the target domain to be more similar to the source domain.

The adversarial loss of the feature discriminator D^f^ is: (2)$${ℒ}_{D^{f}}=\frac{1}{M}\sum_{x^{s}\in X^{S}}{ℒ}_{D}\left(P_{x^{s}}^{f},1\right)+\;\;\frac{1}{N}\sum_{x^{t}\in X^{T}}{ℒ}_{D}\left(P_{x^{t}}^{f},0\right),$$
where ${ℒ}_{D}$ is the binary cross-entropy loss, and *M* and *N* are the number of source domain images and target domain images, respectively; $X^{S}$ is the source domain dataset and $X^{T}$ is the target domain dataset; $P_{x}^{f}$ is the predicted fused feature of the image x.

#### 2.2.2. Discriminator on Output Space

Adversarial learning on the feature space ensures the high-level feature distributions of the source domain and target domain are consistent, and encourages the segmentation network to produce a structurally correct prediction. Nevertheless, this is not enough for accurate segmentation. In addition to the high-level semantics, low-level details also have a huge impact on the segmentation predictions. Thus, adversarial learning on the output space is further used to ensure the source-target consistency on the segmentation details.

For the domain adaptation on the output space, we utilize the entropy map discriminator D^e^ to align the entropy map calculated by the pixel-wise mask. The entropy is the expected value that measures the uncertainty of the predicted mask ($p_{x}^{m})$. We calculate the entropy map with the formula:(3)$$E_{x}=p_{x}^{m}\cdot \log\left(p_{x}^{m}\right).$$

As a rule of thumb, the prediction on the target domain is more uncertain than the prediction on the source domain. In other words, the segmentation network optimized by the source samples would perform better on the source domain. Therefore, we enforce entropy maps of the target domain to approximate that of the source domain. Similarly to the feature discriminator, the alignment on the output space is completed when the entropy maps from the target domain and source domain are difficult to distinguish from each other. The loss function of the entropy map discriminator D^e^ is:(4)$${ℒ}_{D^{e}}=\frac{1}{M}\underset{x^{s}\in X^{S}}{\sum }{ℒ}_{D}\left(E_{x^{s}},1\right)+\;\;\frac{1}{N}\underset{x^{t}\in X^{T}}{\sum }{ℒ}_{D}\left(E_{x^{t}},0\right)$$

### 2.3. Training Procedure and the Overall Loss Function

We use source domain images ($X^{S}$) and labels ($Y^{S}$) and target domain images ($X^{T}$) as training data. The images from both the source domain and the target domain are input into the segmentation module ASDB to gain the predicted masks and boundaries. The ASDB is optimized with the ${ℒ}_{seg}$ between the prediction $P_{X^{S}}$ on the source domain images and manual annotations $Y^{S}$ (Equation (1)). Multi-space discriminators distinguish the entropy maps and features from two domains, respectively. Two discriminators are optimized independently.

The segmentation network and multi-space discriminators achieve the alignment of the source and target domain through adversarial learning. The segmentation network generates fused features and entropy maps for both of the domain datasets. Multi-space discriminators need to identify the source domain membership as true and the target domain membership as false. For the whole framework, the adversarial loss of D^f^ and D^e^ are:(5)$${ℒ}_{adv}^{f}=\frac{1}{N}\sum_{x^{t}\in X^{T}}{ℒ}_{D}\left(P_{x^{t}}^{f},1\right),$$
(6)$${ℒ}_{adv}^{e}=\frac{1}{N}\sum_{x^{t}\in X^{T}}{ℒ}_{D}\left(E_{x^{t}},1\right).$$

The minimum of the adversarial loss renders the segmentation prediction on the target domain and the source domain indiscernible, and more similar features and entropy maps force discriminators to improve their discriminating ability. The ASDB and discriminators are optimized in turn for the domain adaptation at the model level and finally complete the segmentation task in the target domain.

The overall loss function of the MACS is:(7)$${ℒ}_{total}={ℒ}_{seg}+\lambda \left({ℒ}_{adv}^{f}+\;{ℒ}_{adv}^{e}\right),$$
where λ is the variadic parameter for the function balance.

## 3. Experiments

### 3.1. Dataset

We evaluated the MACS on two echocardiography datasets, acquired with different ultrasound equipment.

The public dataset CAMUS [28], collected from 450 patients, included 1800 frames of cardiac ultrasound images with annotations. The apical two-chamber (A2 C) view and apical four-chamber (A4 C) view were acquired by GE© M5 S for each patient. Three cardiologists provided and revised the manual contour of the LV_endo_ and LV_epi_ in the end-diastole (ED) and end-systole (ES). 

Our self-acquired echocardiography dataset was collected from 11 subjects by Vinno© G86. The size of each image was 980 × 650 pixels. Each patient provided sequence images of multiple cardiac cycles in the A2 C view and A4 C view. The dataset was composed of images, sampled every ten frames apart from the echocardiographic sequence. We used the same contouring protocol as the CAMUS for labeling. A cardiologist labeled these images. We utilized 900 unlabeled images and 477 annotated images of this dataset to complete the following experiment. 

### 3.2. Implementation Details

The images were interpolated or cropped to the equal length and width and resized to 256 × 256 pixels for the memory consumption reduction. Data augmentation was accomplished by random crop, adding salt pepper noise, and adjusting intensity.

We implemented the network based on PyTorch (version 1.9.1) and trained the model on two 12 GB Nvidia© TITAN Xp GPUs. The discriminator D^e^ and D^f^ were optimized with the SGD optimizer. The SGD optimizer applied an initial learning rate of 10^−3^ and was divided by 0.2 every 100 epochs. The segmentation network ASDB was optimized with the Adam optimizer. The initial learning rate of Adam was 2.5 × 10^−5^. The batch size of the model was 8 and the network was trained for 200 epochs without warmup epochs. Validation ran every 10 epochs for the model selection by mean dice index.

### 3.3. Evaluation Metrics

We quantitatively evaluated the segmentation performance on echocardiography through the Hausdorff Distance (HD), the Dice index (DI) and the Jaccard index (JI). The HD is a metric based on the boundary of the segmentation prediction, and the DI and JI are region-based segmentation metrics.

The criteria are defined as:(8)$$\mathrm{HD}=\;\max\{\;\underset{x\epsilon X}{\max}\underset{y\epsilon Y}{\min}\|P_{x}^{m}-y\|,\underset{y\epsilon Y}{\max}\underset{x\epsilon X}{\min}\|y-P_{x}^{m}\|\},$$
(9)$$D\;I=\frac{2\times N_{TP}}{2\times N_{TP}+N_{FP}+N_{FN}}$$
(10)$$\mathrm{JI}\;=\frac{N_{TP}}{N_{TP}+N_{FP}+N_{FN}},$$
where $N_{TP},\;N_{FP}$, and $N_{FN}$ are the number of true positive, false positive, and false negative pixels of the $P_{X}^{m}$, respectively, and Y is the ground truth of segmentation.

## 4. Result

In this section, we conduct a set of experiments on the CAMUS dataset and our self-acquired dataset to evaluate the MACS. The ablation study in Section 4.1 analyzes the importance of each component of the proposed model. Then, we alter the number of unlabeled target domain images that participate in the training model and explore the impact in Section 4.2. In Section 4.3, the MACS is compared with groups of domain adaptation and segmentation networks to demonstrate the superiority of the joint model. Section 4.4 presents the comparison with joint adversarial learning methods, which displays the effectiveness of our model. The further experiment in Section 4.5 confirms that our cross-domain segmentation model can achieve competitive accuracy.

### 4.1. Ablation Study

The ablation study is conducted to confirm the effectiveness of the key modules of the MACS we proposed. Specifically, we use the four following modules to segment the LV_endo_ and LV_epi_: ASDB, *w/o* Domain Adaptation (DA);ASDB + Feature discriminator, *w/o* Entropy Domain Adaptation (EDA);ASDB + Entropy discriminator, *w/o* Feature Domain Adaptation (FDA);Our proposed method (MACS).

In the ablation study, the self-acquired dataset and the CAMUS dataset are defined as the source and the target domain, respectively.

Figure 4 displays the representative LV_endo_ and LV_epi_ segmentation predictions on the CAMUS. The ASDB can recognize the ROI of the cross-domain segmentation task, but the prediction masks are not ideal. The segmentation predictions of the ASDB without any discriminator have a large structural error. Either a single discriminator in the feature space or the output space corrects the structural mistake and encourages the segmentation network to obtain accurate results. The integrated MACS with multi-space discriminators further improves the segmentation performance of the boundaries and achieves the best segmentation result of all of these modules.

Table 1 demonstrates the quantitative results of the segmentation performance with different modules. The addition of discriminators improves the mean DI by 2.2% for the segmentation result in the CAMUS dataset.

For the full MACS model, the average running time of segmenting LV_endo_ and LV_epi_ on a single echocardiography is approximately 0.175 s.

### 4.2. Impactation of Target Domain Images’ Number

Various numbers of target domain images may affect the effect of the multi-space domain adaptation of the model and affect the segmentation on the target domain. To explore the causality between these two, we train the framework with different quantities of target domain images. In this experiment, the CAMUS dataset is the source domain dataset and the self-acquired dataset is the target domain dataset. Considering that there are a total of 1600 source domain images taking part in the training, we randomly sample 200, 400 and 800 unlabeled images from the target domain for the rate of 1:8, 1:4 and 1:2 between the target and source domain. In addition, we use 100 unlabeled target domain images to represent the situation of a small amount of data for the adaptation in order to perfect the experiment. 

According to Table 2, with the increase in the number of target domain images that take part in the training process, the DI and the JI of both the LV_endo_ and LV_epi_ are improved. After the number of target domain images reaches more than 400, the accuracy of the segmentation tends to plateau.

The trends of the segmentation DI are visualized in Figure 5. With the increase in the number of target domain images for training the model, the average of the mean DI and the upper limit of the mean DI grows, while the median of the mean DI does not change significantly.

### 4.3. Comparison with Unsupervised Domain Adaptation Methods

We compare our MACS with the state-of-the-art unsupervised domain adaptation approaches: SAGAN [17], CUT [19], and F-LSeSim [18]. The image-to-image translation is from the self-acquired dataset to the CAMUS dataset. The self-acquired dataset A4 C view images are translated to the target domain and named X^T→S^, which maintain the intrinsic content while having the style of source domain images. Then, the DAF is utilized to segment the X^T→S^. 

The image-to-image translation results and segmentation predictions are presented in Figure 6. The CUT and F-LSeSim translate the echocardiography without a noticeable deformation, but the details have been confused with the noise, which causes the failed segmentation. The segmentation predictions on the images translated by the SAGAN are visually similar to the ground truth, but there is a great deformation between the fake image and the original image. The doctor cannot accept the result obtained from such a distorted image. Thus, these domain adaptation methods cannot be applied to solve the problem of cross-domain segmentation in echocardiography. 

The quantitative results from the comparative experiment are reported in Table 3. The MACS reduces the mean HD by at least 3% compared with the segmentation after the image-to-image translation. The mean DI and JI are also significantly improved.

### 4.4. Comparison with Joint Adversarial Learning Models

To confirm the effectiveness of our method, the MACS is compared with the state-of-the-art adversarial learning methods: p-SOAL [21] and BEAL [22]. Considering the area occupied by the left ventricle in the echocardiography is not exiguous, we remove the ROI extraction module of p-SOAL. All of the adversarial learning models are trained with the CAMUS training set and 400 unlabeled images from the self-acquired dataset, and are then tested on 384 rest images of the self-acquired dataset. Table 4 shows the segmentation results of the quantification. Comparing the result from the simple U-Net, all of the adversarial learning methods improve the generalizability of the segmentation network. Our MACS performs better on LV_endo_ segmentation than the other two adversarial learning methods.

Figure 7 presents the Bland-Altman analysis of our MACS and other adversarial learning methods on the segmentation results of the self-acquired dataset. The Bland-Altman plots reflect the consistency of the segmentation results and the ground truth. In each subplot, the forest green or royal blue line indicates the average bias of the organ area, and the brown lines indicate 95% confidence intervals (CI) of the area bias. For our method, the average bias of the LV_endo_ and LV_epi_ areas are 0.45 and 2.8 compared with the ground truth. The plots show that the average bias of the MACS is less than the other two methods and the segmentation for the LV_endo_ has a higher performance than for the LV_epi_. Figure 8 displays the representative segmentation performance on the self-acquired dataset.

### 4.5. Comparison with SOTA Supervised Methods on the CAMUS Dataset

To show a more intuitionistic opinion of the performance, all 477 annotated images in the self-acquired dataset are used for training the MACS, and the well-trained model is tested on the CAMUS testing set. The segmentation results are compared with those of some state-of-the-art fully-supervised methods on the CAMUS and the inter/intra -observer variability, offered in [28]. These supervised methods are trained by 1600 images with labels in the CAMUS. We show the results in Table 5.

Our method achieves a segmentation accuracy comparable to the intra-observer variability; this is particularly true for the segmentation of the LV_endo_ as the DI of the MACS segmentation results exceeds the inter-observer discrepancy by 2.5%. However, the segmentation prediction still has a gap with the inter-observer variability. In addition, we can observe that some of the existing state-of-the-art methods have also not attained a performance higher than the inter-observer on the CAMUS. Compared with the state-of-the-art supervised models, our MACS is trained with fewer annotated images and does not use labels of the public dataset at all.

## 5. Discussion

In this work, we proposed a multi-space adaptation-segmentation-joint framework, named the MACS, for generating an accurate segmentation prediction of the unseen target domain. It addresses the challenge of enlarging the model generalizability when the domain shifts. With the domain adaptation in the feature space and the output space, the MACS trained by the source domain dataset generalizes to the target domain dataset.

Current echocardiography segmentation methods mainly focus on a single dataset. However, imaging devices and medical centers affect the gap between different datasets. As a result, the current segmentation models trained by one echocardiography dataset cannot be generalized to another dataset. For a single medical center, physicians cannot apply a model trained by other center’s datasets to their own images, they have to invest a lot of time labeling their data and retraining the model. This reduces the usability of automated segmentation methods in real clinical diagnosis.

The image-to-image translation is a commonly used strategy for cross-center tasks, which can visually bridge the gap between the source and target domain. However, as shown in Figure 6, image-to-image translation methods cannot fully preserve the features and content concerned with the segmentation task when generating medical images. The loss of the segmentation-related information greatly influence the following task. Therefore, directly adapting the domain by image-to-image translation is not able to solve the cross-domain segmentation in echocardiography. Our MACS avoids the interference of the information loss via the domain adaptation on the intermediate feature space and output space. The adversarial learning strategy allows the model to learn the segmentation features from the original images. The model can heed the segmentation-related detail of images during the adaptation and generalizes the segmentation performance to the target domain. According to Table 3, the joint framework performs significantly better than the segmentation operation after the domain adaptation is conducted separately.

The MACS improves the segmentation performance in the target domain via the adversarial learning between the segmentation network and discriminators. As demonstrated in Table 5, the MACS trained by the self-acquired echocardiography dataset attains the mean DI of 0.89 on the CAMUS dataset, which is better than the inter-observer segmentation consistency offered. It is worth stressing that the MACS achieves such accuracy without the use of the manual label of the CAMUS dataset, while other supervised methods are required to be trained with thousands of annotated images. The results show that the MACS can avoid many of the labeling costs in clinic and create a premise for automatic echocardiography segmentation in clinical applications. The MACS is a practical method for aiding clinical diagnosis.

Although the MACS has improved the segmentation performance on unseen domain echocardiography, there are still limitations waiting to be consummated. The feature discriminator and the entropy map discriminator are optimized independently during adversarial learning, while the high-level fused feature and the segmentation mask are not independent of each other. The relation between them may also affect the adaptation of the model. In the future, the more effective strategy will be explored to collectively optimize two discriminators to improve the segmentation performance of the model.

## 6. Conclusions

We present a segmentation framework named the MACS for LV_endo_ and LV_epi_ segmentation in echocardiography from different domains. The domain adaptation on both the feature space and output space in the model assists in maintaining the performance of the segmentation network on unseen domain images. Concretely, the model extracts fused features and entropy maps from different domains, then aligns them with adversarial learning. The effectiveness of each component is appraised through the ablation study. Our method outperforms the state-of-the-art methods on two datasets, which verifies the elegance and accuracy of the MACS on cross-domain echocardiography segmentation. Considering our self-acquired dataset is not able to cover various cardiac pathological situations, future research will evaluate the model on a larger dataset.

## Figures and Tables

**Figure 1 sensors-23-01479-f001:**
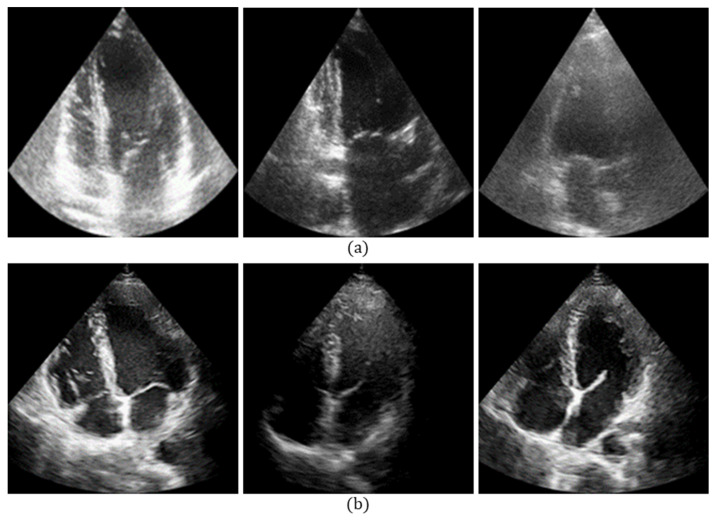
Echocardiography samples from different medical centers and vendors (**a**) CAMUS dataset: GE Vivid E95 ultrasound scanners with a GE M5 S probe and (**b**) self-acquired dataset: Vinno G86 ultrasound scanners with a S1–6 PX probe.

**Figure 2 sensors-23-01479-f002:**
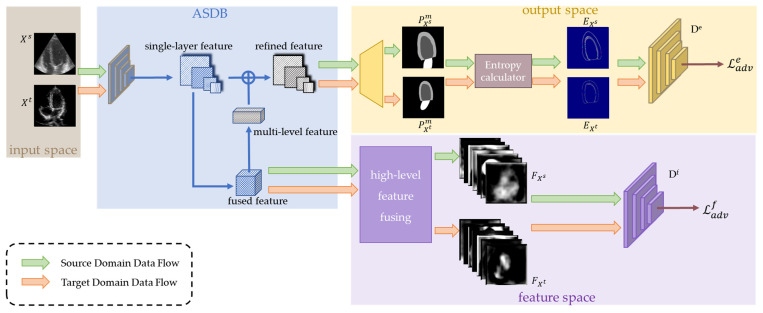
Overview of the proposed MACS. The MACS consists of an attention-based dual-branch segmentation module (ASDB), a discriminator in feature space, and a discriminator in the output space. Source domain images ($X^{S}$) and target domain images ($X^{T}$ ) are segmented by the ASDB which is optimized using only source domain samples. The output of the segmentation network contains predicted masks ($P_{X}^{m}$ ) and high-level intermediate features ($F_{X}$ ). The entropy map $\left(E_{X}\right)$ is calculated from the segmentation prediction $P_{X}^{m}$. and $E_{X}$ from two different domains are forced to be aligned by discriminators, making the ASDB output accurate segmentation on unseen target domain dataset.

**Figure 3 sensors-23-01479-f003:**
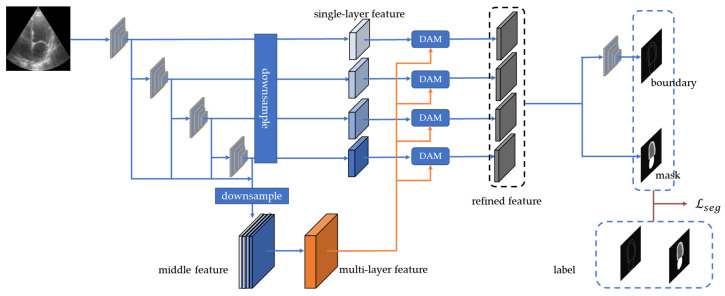
The architecture of the Attention Segmentation with the Dual-Branch (ASDB) module. The encoder extracts multiply single-layer feature maps. The multi-layer feature is the fusion of these feature maps. Single-layer features and multi-layer feature pass by the deep attention model (DAM) for the refined feature. The dual-branch structure generates the prediction of the mask and boundary.

**Figure 4 sensors-23-01479-f004:**
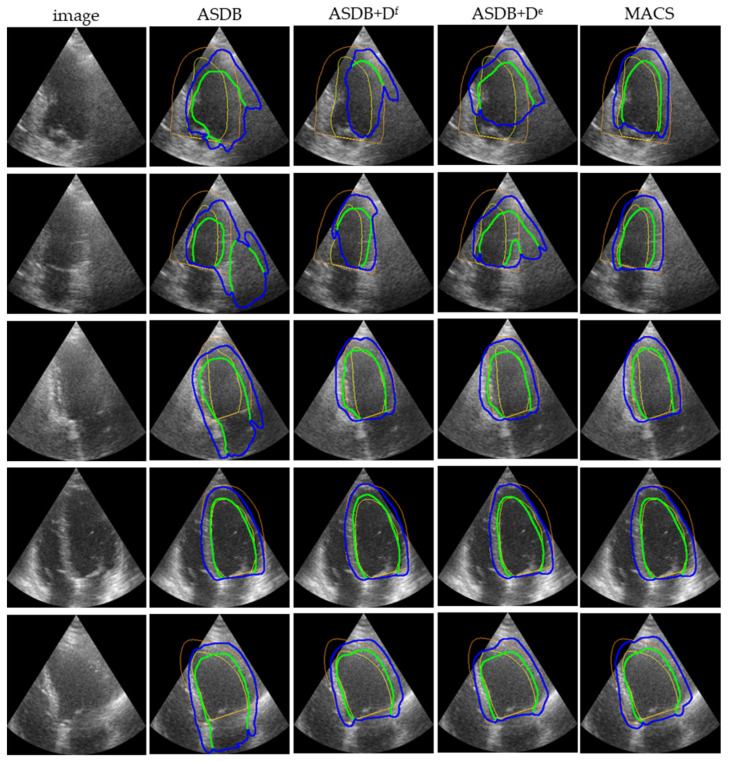
Representative visual segmentation results of the ablation study. Different model configurations are compared, including the ASDB, the ASDB + Feature discriminator (D^f^), the ASDB + Entropy discriminator (D^e^), and the full framework of the MACS. The green and blue contours indicate boundaries of the LV_endo_ and LV_epi_, respectively. The yellow and orange colors denote the ground truth.

**Figure 5 sensors-23-01479-f005:**
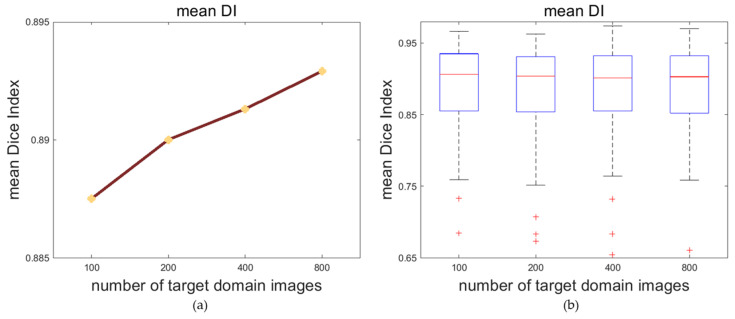
The mean Dice Index (DI) for segmentation results of the MACS trained with different numbers of target domain images. (**a**) The line graph of the mean DI. (**b**) The boxplot of the mean DI.

**Figure 6 sensors-23-01479-f006:**
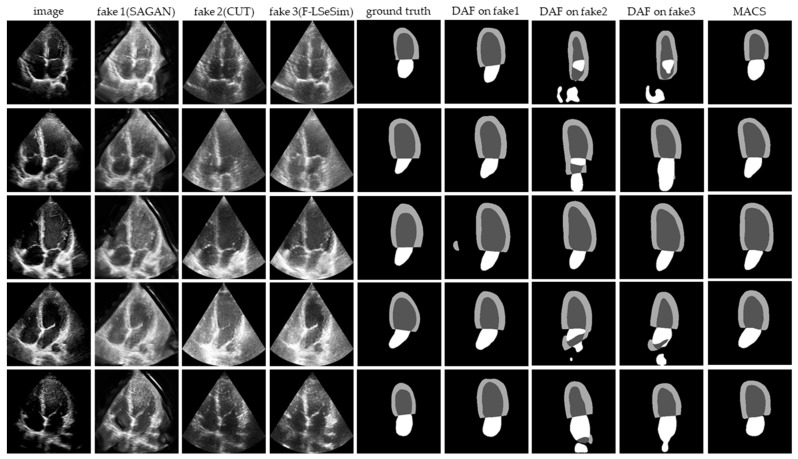
Fake images generated by image-to-image translation methods (SAGAN, CUT, F-LSaSim) and the visualization prediction of fake images that segmented by the DAF.

**Figure 7 sensors-23-01479-f007:**
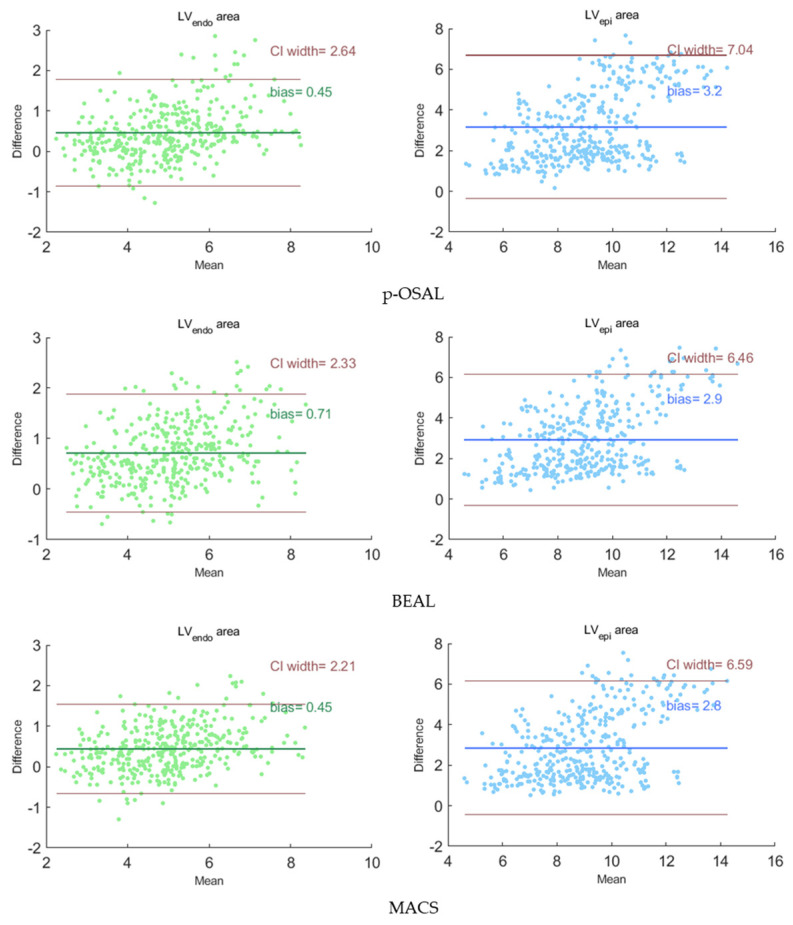
The Bland-Altman analysis on adversarial learning methods about the organ area (×10^3^ pixel^2^). The x-axis represents the average area of the prediction and the ground truth (GT). The y-axis represents the bias of the area of the prediction and the GT. The forest green or royal blue line indicates the bias of the organ area and brown lines indicate 95% confidence intervals of the area bias.

**Figure 8 sensors-23-01479-f008:**
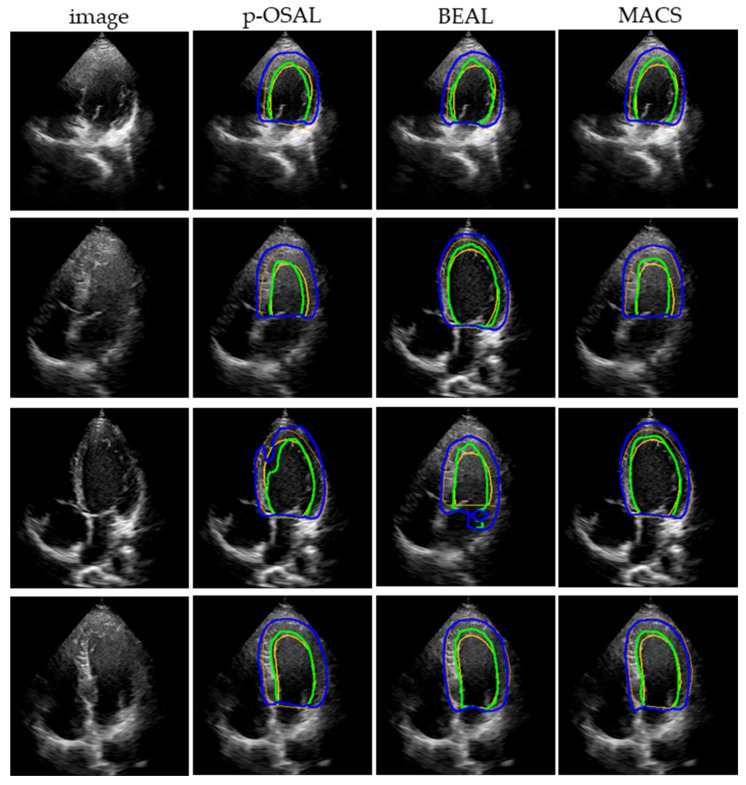
Representative visual segmentation results of adversarial learning methods. Green and blue contours indicate boundaries of the LV_endo_ and LV_epi_, respectively. Yellow and orange colors denote the ground truth.

**Table 1 sensors-23-01479-t001:** Ablation studies on the proposed MACS. The fully domain adaptation (DA), entropy domain adaptation (EDA) and feature domain adaptation (FDA) are ablated, respectively. The up arrow indicates expecting a higher value.

Methods		HD (↓)	DI(↑)	JI(↑)
		mm	val.	val.
*w/o* DA	LV_endo_	6.38	0.8649	0.7738
	LV_epi_	7.94	0.8777	0.7889
*w/o* EDA	LV_endo_	6.41	0.8630	0.7680
	LV_epi_	7.70	0.9038	0.8278
*w/o* FDA	LV_endo_	6.56	0.8637	0.7700
	LV_epi_	7.73	0.8949	0.8138
MACS	LV_endo_	6.30	0.8764	0.7881
	LV_epi_	7.59	0.9037	0.8263

**Table 2 sensors-23-01479-t002:** Comparison of our MACS model trained with different numbers of target domain images. The up arrow indicates expecting a higher value.

	HD (↓)	DI(↑)	JI(↑)	Target Images Number
	mm	val.	val.	
LV_endo_	5.70	0.8983	0.8183	800
LV_epi_	7.33	0.8874	0.7996	
LV_endo_	5.65	0.9033	0.8265	400
LV_epi_	7.64	0.8792	0.7910	
LV_endo_	5.72	0.8967	0.8160	200
LV_epi_	7.45	0.8832	0.7973	
LV_endo_	5.69	0.8926	0.8094	100
LV_epi_	7.41	0.8824	0.7950	

**Table 3 sensors-23-01479-t003:** Comparison with the segmentation performance on images translated by the SAGAN, F-LSeSim and CUT. The up arrow indicates expecting a higher value.

Methods	LV_endo_			LV_epi_		
	HD (↓)	DI (↑)	JI (↑)	HD (↓)	DI (↑)	JI (↑)
	mm	val.	val.	mm	val.	val.
SAGAN	6.03	0.8690	0.7715	7.28	0.9174	0.8419
F-LSeSim	6.60	0.8264	0.7125	7.93	0.8669	0.7707
CUT	7.01	0.7571	0.6236	8.34	0.8171	0.7020
MACS	5.79	0.8847	0.7963	7.11	0.9167	0.8475

**Table 4 sensors-23-01479-t004:** Comparison with SOTA adversarial learning methods. The up arrow indicates expecting a higher value.

Methods		HD (↓)	DI (↑)	JI (↑)
		mm	val.	val.
U-Net	LV_endo_	6.47	0.8457	0.7462
	LV_epi_	8.10	0.8630	0.7675
p-OSAL	LV_endo_	5.68	0.8982	0.8178
	LV_epi_	7.68	0.8734	0.7820
BEAL	LV_endo_	5.72	0.9030	0.8260
	LV_epi_	7.29	0.8865	0.8021
MACS	LV_endo_	5.65	0.9033	0.8265
	LV_epi_	7.64	0.8792	0.7910

**Table 5 sensors-23-01479-t005:** Comparison with SOTA supervised methods on the CAMUS dataset. The up arrow indicates expecting a higher value.

Methods		HD (↓)	DI (↑)	JI (↑)
		mm	val.	val.
Inter-obs	LV_endo_	8.50	0.8545	—
	LV_epi_	6.45	0.9035	—
Intra-obs	LV_endo_	4.55	0.9375	—
	LV_epi_	5.00	0.9540	—
DAPIS	mean	5.26	0.9240	—
CLAS	LV_endo_	—	0.9130	—
	LV_epi_	—	0.9400	—
TRSA-Net	LV_endo_	0.67	0.9543	—
	LV_epi_	1.33	0.8678	—
DAF	LV_endo_	5.07	0.9335	0.8774
	LV_epi_	6.44	0.9549	0.9143
MACS	LV_endo_	6.30	0.8764	0.7881
	LV_epi_	7.59	0.9037	0.8263

## Data Availability

The authors do not have permission to share data.

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
