# Peer review of "Cross-Domain Echocardiography Segmentation with Multi-Space Joint Adaptation"

_sensors, 2023, doi:10.3390/s23031479_

Round 1

Reviewer 1 Report

The issues are:

 -Abstract is not able to convey what is the technical contribution of this paper. I suggest to re-write it.
-Improve the quality of figures and explain those properly.
-The result section is weak and I suggest authors add more results and compare those with the existing approaches.
-Although this paper is well written, there are still some typos in the current version.

-The review of the state of the art lacks an analysis of the existing work, Also I suggest a few good papers that help to improve the literature work. The papers are: A survey of deep active learning, Synergic deep learning for smart health diagnosis of COVID-19 for connected living and smart cities, Joint computation offloading and tsk caching for multi-user and multi-task MEC systems: reinforcement learning-based algorithms, Four-image encryption scheme based on quaternion Fresnel transform, chaos and computer generated hologram
-The authors are expected to report the running time of the proposed algorithm in the revision.
-As an additional remark, references need to be completed with all the required information (e.g. page number, name of journal/conference, vol., issues, etc).

Reviewer 2 Report

Authors aim to evaluated the MACS method on two echo-cardiography datasets from different medical centers and vendors, the publicly available CAMUS dataset and their self-acquired dataset. Overall the papere is well written, and few adjustments should be conducted.

1. Authors should revise the part of the introduction in which they elaborate the papers' contributions (Lines 104-116). Instead of the bullet points, the contributions should be elaborated in one pararagraph. Each of the contributions should be supported at least with one reference. 

2. Paragraph describing the paper content should be added at the end of the Introduction.

3. There should be summary subchapter at the end of the Result chapter with the summarized selected comparison of the proposed method in one table, with the comments. It is hard to grasp the overall comparison as it is currently described in various chapters

4. The current version of the discusssion and conclusion is hard to follow. Conclusion should be expanded in the following manner, and possibly merged with the discussion section. 

(1).     Summary of the research - what was the goal, and how was it attained (2).     Theorethical implications - Discussion of why the authors found these results and how they comply (or not) with the Literature Review. (3).     Managerial Implications (4).     Limitations of the paper (5).     Future Studies and Recommendations    5. Authors compare the public dataset with the large number of patients, with their own data on only 11 patients. This is a relevant limitation of the paper, which is not addressed in the conclusion.  If you want, you can merge the current text in the discussion section with the new conclusion (e.g. use parts of the text). 
